# Optimal Subsampling with Influence Functions

**Daniel Ting**
Tableau Software
Seattle, WA, USA
dting@tableau.com

**Eric Brochu**
Tableau Software
Vancouver, BC, Canada
ebrochu@tableau.com

## Abstract

Subsampling is a common and often effective method to deal with the computational challenges of large datasets. However, for most statistical models, there is no well-motivated approach for drawing a non-uniform subsample. We show that the concept of an asymptotically linear estimator and the associated influence function leads to asymptotically optimal sampling probabilities for a wide class of popular models. This is the only tight optimality result for subsampling we are aware of as other methods only provide probabilistic error bounds or optimal rates. We also show that these optimal weights can differ depending on whether the task is parameter estimation or prediction. Furthermore, for linear regression models, which have well-studied procedures for non-uniform subsampling, we empirically show our optimal influence function based method outperforms previous approaches even when using approximations to the optimal probabilities.

## 1   Introduction

As the amount of data increases, the question arises as to how best to deal with the large datasets. While computational platforms such as Spark [28] and Ray [23] help process large datasets once a desired model is chosen, simply using smaller data can be a faster solution for exploratory data modeling, rapid prototyping, or other tasks where the accuracy obtainable from the full dataset is not needed. Sampling provides a flexible summarization of the data that can be applied to almost all tasks in a simple, straightforward manner.

Ideally, data is sampled *efficiently*, preferentially sampling the data that will accurately approximate the estimates from the full data set. However, we show existing preferential sampling techniques are inefficient and can demonstrate pathological behavior. Furthermore, most methods are inflexible as they are derived only for specific linear or logistic regression models.

We propose using the *influence function* as a measure of sampling importance. The influence function measures the change in an objective or values of interest due to a single point. It is a particularly general approach as many model and estimators, such as maximum likelihood and M-estimators, can be cast in the framework, and it can work with non-differentiable objectives. We prove that the regularized version of our sampling design is asymptotically optimal among all regularized designs of the same expected size. While other sampling methods provide probabilistic guarantees on the resulting samples, we do not know of other results that can claim optimality.

Beyond the improved performance of our method, the influence-based approach allows one to understand the problem of optimal subsampling. The influence function and the notion of asymptotically linear estimators reduces the problem of accurately approximating an estimate on the full data set to a problem of calculating the mean over influence functions. Thus, the problem of finding an optimal sampling probabilities, or *sampling design*, for estimating a model is converted to the more straightforward problem of generating an optimal sampling design for a mean. This design can depend on the specific task on hand. In particular, efficient sampling designs to estimate the parameters of a model

---
**Algorithm 1** Basic Influence Based Sampling$(X, Y)$

    Given a model and objective, derive the influence function $\psi_\theta$ or an approximation of it

    Compute a pilot estimate of the parameters $\hat{\theta}$.

    Draw an importance sample taking $(X_i, Y_i)$ with probability $\pi_i \propto \|\psi_{\hat{\theta}}(X_i, Y_i)\|$ under the constraint $\pi_i \geq \alpha$

    Fit a model with the sample using importance weights $1/\pi_i$

---

can substantially differ from efficient sampling designs for predictions even when both use the exact same model. This fact is also borne out in our experimental results. We explicitly derive sampling probabilities for linear regression, quantile regression, and generalized linear models (GLMs). In these examples, the "influence" of the residuals $y_i - \hat{y}_i$ may be separated from the "influence" of the regression design or predictors $X$. As a result, existing approaches are shown to only exploit one of the two, whereas our method appropriately incorporates both.

## 1.1 Related work

The dominant subsampling approach in the literature for least squares regression is based on statistical leverage scores [10, 19]. A number of papers [4, 5, 8, 21] address more general $L_p$ linear regression problems and derive a corresponding leverage for $L_p$ regression. These methods focus on generating sampling designs from the matrix of predictors $X$, and make no or limited use of the responses $Y$. The resulting sampling designs are obtained via a relatively expensive to compute random projection or low distortion embedding. The sampling weights for leverage based sampling are defined by

$$w_i^{(\text{lev})} = H_{ii}, \quad \text{where } \hat{Y} = HY = XX^\dagger Y \tag{1}$$

with $\dagger$ indicating the Moore-Penrose pseudo-inverse, so that $H$ is the hat matrix.

The gradient-based approach of [29] for linear models and local case-control sampling [13] for logistic models generate sampling probabilities based on the residuals given a pilot estimate of the coefficients. Very recent work covers optimal subsampling for logistic regression [27]. The gradient based approach is similar to ours in that it provides a general method that can be applied across a wide range of models. Given a differentiable M-estimator objective $J(\theta) = \sum_i \ell(\theta; x_i)$, gradient-based sampling defines importance sampling weights

$$w_i^{(\text{grad})} = \|\nabla \ell(\theta; x_i)\|. \tag{2}$$

Other techniques such as SGD [3], the Uluru algorithm [9] and random projection techniques [22] are relevant for fast model fitting on large datasets. In these cases, the data size is not reduced, and the result is an estimate for the specific model being fit. In contrast, we focus on sampling as a way to reduce computational complexity for one or more models while being able to use the sample for additional purposes, for example, coarse parameter tuning or rapid prototyping on a single machine.

Coreset generation [1, 15] is another technique for generating reduced size datasets that can well approximate results generated from fitting a model on a full dataset. These techniques provide a probabilistic guarantee on the quality of the approximation according to some pre-specified sensitivity and cost metrics and some strong assumptions on the solution. In earlier literature, similar techniques without specific guarantees were also known as data squashing [11, 20]. We consider our work to be complementary to the coreset literature. Our work shows that the influence function can be used to improve the sensitivity measure used in many coreset designs. For example, [15] defines a sensitivity metric on the log-likelihood for logistic regression while our work shows that choice is inefficient, but that appropriately scaled derivatives of the log-likelihood define an asymptotically optimal metric.

We also note the connection — or lack thereof — between sampling for statistical purposes and subsampling for computational purposes. While the former is a fundamental topic in statistics, the latter discards data, which is typically contrary a statistician's goals. In sampling, the population data is unknown, and cost arises from a data collection process. In contrast, the data is known in subsampling and can be exploited to draw better samples. Size is the main cost in this case. Though there is limited work on subsampling procedures in the statistics community, several papers have analyzed statistical properties of these procedures [19, 24].

## 2 Our method

Our key idea is that many parameter estimators can be asymptotically expressed as a mean of influence vectors. This allows the problem of optimal subsampling for a statistical model to be recast in terms of the well-studied problem of optimal subsampling for a mean. We give a brief overview of influence functions and asymptotically linear estimators. The interested reader may refer to [26] and [14] for more information. We then show how this technique can generate sampling weights. As the exact influence function is often expensive to compute, we address practical considerations in implementing our method. Finally, we analyze the asymptotic error of our method and show that out of all importance sampling distributions satisfying a form of regularization, ours has the lowest asymptotic variance. We summarize our method in Algorithm 1.

### 2.1 Influence and asymptotically linear estimators

We consider the class of plug-in estimators $\hat{\theta}(P)$ taking a distribution to a real-valued vector of parameter estimates. This is a highly flexible class of estimators. For example, any M-estimator $\hat{\theta}(P) = \operatorname{argmin}_\theta \ \mathbb{E}_P \ell(\theta, X)$ is of this form for loss $\ell$. The argument for the M-estimator is the empirical distribution $\mathbb{P}_n$. When it exists, the influence function for this estimator is defined by its functional Gâteaux derivative

$$\psi_\theta(x) = \lim_{\epsilon \to 0^+} \frac{1}{\epsilon} \left( \hat{\theta}\left( (1-\epsilon)P_\theta + \epsilon \delta_x \right) - \hat{\theta}(P_\theta) \right) \tag{3}$$

where $\delta_x$ is the Dirac delta measure at $x$. It represents the infinitesimal change in the estimate by adding the point $x$ to the sample. The estimator $\hat{\theta}$ is an *asymptotically linear estimator* with influence function $\psi_\theta$ if it satisfies

$$\sqrt{n} \left( \hat{\theta} - \theta \right) = \frac{1}{\sqrt{n}} \sum_{i=1}^{n} \psi_\theta(X_i) + o_p(1), \tag{4}$$

with $\mathbb{E}\psi_\theta = 0$, $\mathbb{E}\psi_\theta^T \psi_\theta < \infty$. Here, $o_p(1)$ denotes convergence in probability in some normed space.

Asymptotically linear estimators are pervasive in statistical modeling. Under sufficient regularity conditions, M-estimators, maximum likelihood estimators, Z-estimators, non-degenerate U-statistics, and Generalized Method of Moments estimators are all examples of asymptotically linear estimators. For maximum likelihood estimation with correctly specified and sufficiently regular models, the influence function is easily derived. It is the gradient of the log-likelihood, also called the score $s_\theta(x)$, scaled by the inverse Fisher information $I_\theta = \mathbb{E}s_\theta(X)s_\theta(X)^T$.

$$s_\theta(x) = \partial \ell(\theta; x)/\partial \theta, \qquad \psi_\theta = I_\theta^{-1} s_\theta. \tag{5}$$

### 2.2 Optimal sampling design and estimation

It is not obvious how one should choose good sampling probabilities. For example, [11, 15] pick sampling weights to best approximate an objective function. For least squares regression, this is equivalent to sampling with probability proportional to the squared residuals. However, Fig. 1 shows how this can actually perform worse than uniform sampling.

The reason for this poor performance is that accurately estimating the loss itself is different from accurately estimating the actual quantity of interest, the minimizer of the loss. The corresponding optimal sampling probabilities for each task are also different. The theory of asymptotically linear estimators addresses the problem of mapping the quantity of interest to optimal sampling probabilities in two ways. Influence functions measure how much a data point changes the actual quantity of interest, in other words, its importance for the task at hand. An asymptotic linear expansion into independent, zero mean influence vectors then reduces the problem of drawing an optimal sample for an estimator to the problem of drawing an optimal sample for a multivariate mean. We further reduce this to a well-studied problem of designing optimal sampling probabilities for a univariate mean.

First, we describe optimal independent (Poisson) sampling for a univariate mean. Let independent $Z_i \sim Bernoulli(\pi_i)$ indicate if $x_i \in \mathbb{R}$ is included in the sample. An optimal design with expected

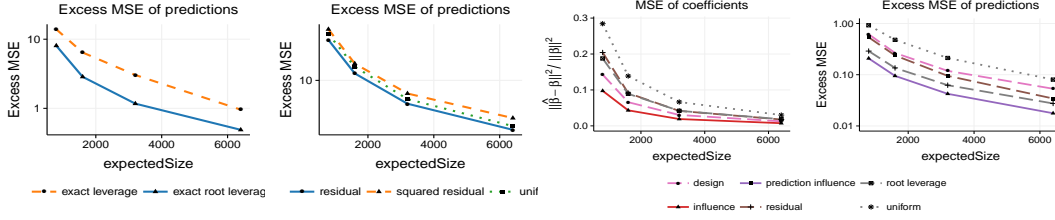

Figure 1: Left: Naive sampling weights versus influence function derived weights on Gaussian data where 5% of points are modified to have high leverage. Root leverage outperforms leverage as predicted by theory. Squared residuals as weights optimally approximate the squared loss but perform worse than uniform weights when approximating the true predictions. Right: The influence based weights are separated in the contributions of the regressors and the residual for the CASP data set. For predictions the contribution of the regressors is the root leverage score. In this case, the regressors are more important than the residual.

size $n_{sub}$ for minimizing the variance is given by

$$\pi^{(\text{opt})} = \underset{\sum \pi_j = n_{sub}}{\operatorname{argmin}} \sum_i \left( \frac{x_i}{\pi_i} \right)^2 \pi_i (1 - \pi_i), \quad \pi_i^{(\text{opt})} = \min\{\lambda |x_i|, 1\} \tag{6}$$

for $\lambda$ such that $\sum_i \pi_i^{(\text{opt})} = n_{sub}$. Such a scheme is a probability proportional to size (PPS) design with respect to the size measure $x_i$.

To reduce the problem of multiple parameter estimation for an asymptotically unbiased and asymptotically linear estimator, consider the trace of the variance

$$\operatorname{Tr}\left( \operatorname{Var}\left( \sum_{i=1}^n \frac{Z_i}{\pi_i} \psi_\theta(x_i) \right) \right) = \sum_{i=1}^n \left( \frac{\|\psi_\theta(x_i)\|}{\pi_i} \right)^2 \pi_i (1 - \pi_i). \tag{7}$$

A PPS design with size equal to the norm of the influence function is optimal for minimizing the trace of the asymptotic variance. This optimality statement is made more precise in section 5.

## 3 Examples

We examine the influence functions for linear least squares regression, logistic regression and GLMs, and quantile regression. While asymptotically linear estimators encompass a wider class of estimators, we focus on this particular subset as we can demonstrate they can be computed efficiently and they share some structure that allows us to understand the problem by decomposing the influence into informative residual and design components.

We also examine the difference between targeting accurate estimation of parameters versus predictions. The form of the weights provides insight on sampling methods by allowing a decomposition into regression design and residual effects.

### 3.1 Linear Least Squares

For linear regression models, the influence function for the coefficients $\hat{\theta}$ is given by

$$\psi_\theta(x_i, y_i) = \left( y_i - x_i^T \theta \right) \Sigma^{-1} x_i \tag{8}$$

where $\Sigma = \frac{1}{n} \left( X^T X \right)$ is the empirical second moment matrix and $\theta$ are the true coefficients. Taking the norm of the influence yields the sampling weight.

This weight differs from gradient-based sampling weights, $\|(y_i - \hat{y}_i)x_i\|$, given in (2) only by the scaling $\Sigma^{-1}$. The influence-based weights appropriate change with scalings of the data while the gradient-based approach displays pathological behavior. A parameter $\theta_j$ is made more important by making it larger, or equivalently, by scaling the $j^{th}$ coordinate of the data to be smaller. Under gradient-based sampling, the opposite occurs; a coordinate is less important as it is scaled to be smaller. As shown in Fig. 2, influence-based sampling correctly increases the importance.

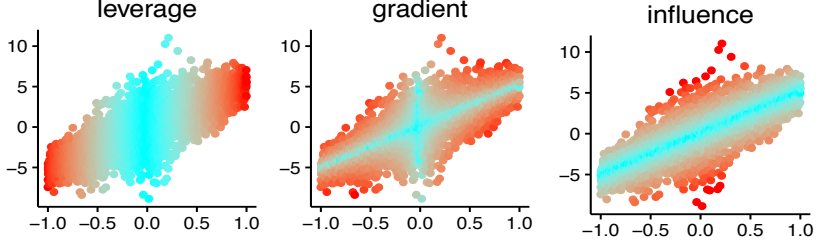

Figure 2: Sampling patterns for different importance measures for the model $Y = 5X + 1 + \epsilon$. Red and light blue respectively indicate high and low inclusion probability. Leverage focuses on the extremes of the regression design. It and the gradient ignore large residuals near the center. The influence picks an appropriate balance of residual and regression design effects.

## 3.2 Influence on predictions

The notion of influence can be extended beyond estimators for parameters. For example, points can be sampled according to their influence on predictions rather than on the coefficients, which is useful when exact coefficient values are unimportant as long as predictive performance is good. The influence on predictions has also been used in other contexts such as in the interpretability of machine learning models [17].

The influence on the prediction is easily derived from the influence on the coefficients. When each prediction $\hat{y}_i(\theta)$ is a twice differentiable function of the $k$-dimensional parameter $\theta$, then, with a slight abuse of notation, the influence on the vector of predictions can be easily computed by the chain rule

$$\psi_\theta^{(\text{pred})}(x_i, y_i) = \hat{y}'(\theta) \frac{d\theta}{d\delta_{(x_i, y_i)}} = \hat{y}'(\theta) \psi_\theta(x_i, y_i). \tag{9}$$

Here $\hat{y}'(\theta)$ is the $n \times k$ matrix of gradients $\nabla \hat{y}_i(\theta)$. In the context of linear least squares regression, the influence is $\psi_\theta^{(\text{pred})}(x_i, y_i) = X(X^TX)^{-1}x_i(y_i - \hat{y}_i) = r_i H_{i\cdot}$ where $r_i$ is the residual of the $i^{th}$ prediction and $H_{i\cdot}$ is the $i^{th}$ row of the hat matrix. Since the hat matrix $H$ is idempotent and symmetric, the squared norm $\|H_{i\cdot}\|^2 = e_i^T H^T H e_i = e_i^T H e_i = H_{ii}$. The prediction influence weight can thus be more succinctly written as $w_i^{(\text{pred})} = |r_i| \sqrt{H_{ii}}$.

The prediction on the influence provides a strong connection to leverage-based sampling. In the classical statistical setting where the experimenter does not have knowledge of the response $Y_i$ while setting the regression design $X$, a sensible measure of influence takes the expectation over the unknown $Y_i$. In this case, one obtains a sampling weight proportional to the root leverage $\sqrt{H_{ii}}$. If only the influence on the single prediction $\hat{y}_i$ is considered, then leverage score based sampling is recovered. The influence is $\psi_{P_\theta}^{(\text{pred,i})}(x_i, y_i) = x_i(X^TX)^{-1}x_i(y_i - \hat{y}_i) = r_i H_{ii}$. Taking the norm and the expectation over an unknown $Y_i$ exactly recovers leverage score sampling. Thus, leverage-based sampling throws away two pieces of available information, the effect of the regression design on points other than the $i^{th}$ point and more importantly, the information about the response $Y_i$.

The surprising result that root leverage can yield a better sampling weight than leverage is borne out in simulations. Fig. 1 shows that root leverage weights yield lower predictive loss than leverage based weights when the model is correctly specified. However, we find the empirical results less clear in the case of real data with misspecified models. In this case, leverage scores can be correlated with the size of the residuals, and leverage sometimes performs better than root leverage.

## 3.3 Generalized Linear Model

GLMs can be solved via iteratively reweighted least squares. The solution is given by $\hat{\theta} = (X^TWX)^{-1}X^TWY$ where $W$ is the diagonal matrix with $W_{ii} = 1/\text{Var}_{\hat{\theta}}(Y_i|x_i^T\hat{\theta})$, the inverse predicted variance at $x_i$ if $\hat{\theta}$ is equal to the true parameters. It is easy to verify that the influence functions for $\hat{\theta}$ and $\hat{y}$ under correct model specification are given by

$$\psi_\theta(x_i, y_i) = (y_i - \hat{y}_i)\left(X^TWX\right)^{-1}x_i, \quad \psi_\theta^{(\text{pred})}(x_i, y_i) = r_i H_{i\cdot}^T. \tag{10}$$

where $r_i = y_i - \hat{y}_i$ is the residual and $H = X(X^TWX)^{-1}X^TW$ so that $\hat{\theta} = HY$. Unlike linear least squares regression, the matrix $H$ is non-symmetric so the norm of the influence function cannot be expressed exactly in terms of the diagonal of the hat matrix.

For the special case of logistic regression, local case-control sampling provides a sampling method that has both good empirical and theoretical properties. It chooses sampling probabilities proportional to the "surprise" $y_i(1 - \hat{p}_i) + (1 - y_i)\hat{p}_i$ so that a point is likely to be sampled only if it did not match the prediction. The surprise can also be expressed as the absolute value of the residual $|y_i - \hat{p}_i|$. Thus, local case-control sampling is equivalent to influence-based sampling under the approximation that there is no effect due to the regression design.

### 3.4 Quantile regression

While linear least squares regression focuses on estimating the conditional mean, in some cases the quantity of interest is not the mean but the median or other quantiles of a distribution. Quantile regression [16] provides another useful generalization of linear models. It is of particular interest here since the objective is non-differentiable but still has an influence function.

For quantile regression, the loss function is the non-differentiable "check" function

$$\ell_\tau(x) = (1 - \tau)x1(x < 0) + \tau x1(x \geq 0)$$

rather than the squared residual. When the desired quantile is $\tau$ and the true conditional quantile is linear, the influence function is given by

$$\psi_\theta(x_i, y_i) = [\tau(1 - \tau)]^{-1}V^{-1}x\,\rho(y_i - x_i^T\theta)$$

where $\rho$ is a subgradient of the loss, $\rho(z) = 1 - \tau$ if $z < 0$ and $\tau$ if $z > 0$, and

$$V = \int xx^Tf(0|x)dG(x)$$

when the $X_i$ are randomly drawn from a distribution $X_i \sim G$ and the error $Y_i - x_i^T\theta$ has density $f(\cdot|x_i)$. In the commonly considered special case where the error distribution is independent of the predictors $X$, $\Sigma = \frac{1}{n}X^TX$ is a consistent estimator of $V$. This gives the following estimated influence functions on the coefficients and predictions

$$\hat{\psi}_\theta(x_i, y_i) = [\tau(1 - \tau)]^{-1}\,\rho(r_i)\Sigma^{-1}x_i, \quad \hat{\psi}_\theta^{(pred)}(x_i, y_i) = [\tau(1 - \tau)]^{-1}\,\rho(r_i)H_{\cdot i}. \tag{11}$$

We note that this influence function has the same form as the influence function for linear regression in (8). The residual $r_i$ in the influence function for linear regression is simply replaced by $\rho(r_i)$ in the quantile influence function. We will refer to $\rho(r_i)$ as the "residual" for quantile regression.

## 4 Implementation

To implement our method practically, we must address three main issues: the fact that the influence function depends on the unknown true parameter $\theta$; the problem that very small sampling probabilities can lead to high variance; and the computational expense of exactly computing the influence function.

**Influence function estimation:** In all the examples, the exact influence function depends on the true parameter $\theta$. A simple solution is to substitute a pilot estimate $\theta_0$ for the true parameter. In some cases, the pilot estimate may be readily available or easily obtained, such as applications in which estimated parameters from one day may be used as a warm-start for training on the next day. If there is no pre-existing pilot estimate, then one can first draw a uniform sample, or even a reasonable convenience sample, from the data to form a pilot that can be used on the remaining data. Theorem 2 shows that simple consistency of the pilot is sufficient to ensure asymptotically optimal estimation of regularized sampling probabilities. It does not need to improve at a rate dependent on the number of points $n$.

**Regularization of probabilities:** It is typical for some estimated influences to be very close to 0. For example, any point that lies on a pilot estimate's regression line has zero influence. Since each sampled point is weighted by its inverse sampling probability, these points have very large weight when selected.

Our solution is to regularize by setting a minimum sampling probability $\alpha$. This corresponds to adding the convex constraint that $\alpha \leq \pi_i \leq 1$ to the optimization in section 2.2. The resulting sampling probabilities are $\pi_i = \max\{\alpha, \min\{1, \lambda\|\psi_{\theta_0}(x_i)\|\}\}$ for $\lambda$ such that $\sum_i \pi_i = n_{sub}$. In other words, the probabilities not equal to $\alpha$ or 1 are drawn with probability proportional to $\|\psi_{\theta_0}(x_i)\|$.

**Influence function computation:** For many estimators, computing the influence function requires a matrix pseudo-inverse for some matrix $\Sigma$ or computing a leverage score. These may be costly to compute. Approximate leverage scores may be computed [10] using fast Johnson-Lindestrauss transforms. For matrix inverse approximations, a standard strategy [3, 2] is to replace $\Sigma$ with its diagonal, which amounts to using the Jacobi preconditioner in place of $\Sigma$ and ignoring covariance.

We apply a slightly more complex approximation and estimate $\Sigma = X^T W X$ with a diagonal and a low rank component so that $\Sigma = S^{1/2}(I - VDV^T)S^{1/2}$ where $V$ has low rank $r$ and $S$ is a diagonal scaling matrix. This can be approximately computed using a truncated SVD of $QXW^{1/2}$ where $Q$ is an $n_{sub} \times d$ sampling matrix or a fast Johnson-Lindestrauss transform. Since the inverse is of interest, the truncated SVD should compute the smallest rather than largest singular values of $XW^{1/2}$. This can be done in $O(n_{sub}dk)$ time where $k$ is the dimension of the data. Computing $\Sigma^{-1}X^T$ takes $O(ndk)$ time.

## 5   Error analysis

Our main theoretical result is the optimality of influence based sampling weights and derivation of its asymptotic variance. To do this, we first define the class of limit importance sampling distributions in terms of Radon-Nikodym derivatives and establish the conditions under which the asymptotic variance for an importance sampling distribution can be derived.

Theorem 1 maps each limit importance sampling distribution to an asymptotic variance. It further shows there is a unique importance sampling distribution that minimizes this asymptotic variance. Our main result, Theorem 2, states that a consistent estimate of the influence function is sufficient to attain this minimum asymptotic variance. The proofs are provided in the supplementary material.

Let $\phi(\cdot)$ be some real-valued function on distributions in some P-Donsker class $\mathcal{F}$. Suppose it is Hadamard differentiable at $P_\theta$ under the uniform norm with influence function $\psi_\theta \in \ell^\infty(\mathcal{F})$. Assume values $X_i$ are drawn i.i.d. from $P_\theta$. Consider the set of measures $Q$ that are mutually absolutely continuous with respect to $P_\theta$ with $\alpha \leq dQ/dP \leq 1$ almost everywhere and the total measure of $Q$ is some constant $c$. Let $\hat{\mathbb{P}}_n^Q = n^{-1}\sum_{i=1}^n Z_i \frac{dP}{dQ}(X_i)\delta_{X_i}$ be the resulting estimated empirical measure where $Z_i$ indicates $X_i$ is in the subsample and equals 1 with probability $\frac{dQ}{dP}$. These statements are easy to understand when $P, Q$ have densities $p, q$ respectively. In this case, the Radon-Nikodym derivative $dQ/dP = q/p$ exists almost everywhere. The subsampling procedure changes the measure from the natural sampling distribution $P$ to the subsample's distribution which is proportional to $Q$. The weights $\frac{p}{q}$ are the familiar importance sampling weights to evaluate an integral over $P$ given a sample from $Q$.

**Theorem 1.** *For any $Q$ that satisfies the assumptions, $\sqrt{\frac{n}{c}}(\phi(\hat{\mathbb{P}}_n^Q) - \phi(P)) \rightsquigarrow \mathcal{N}(0, V^Q)$ where $V^Q = \int \psi(x)\psi(x)^T \left(\frac{dP}{dQ}\right)^2 dQ(x)$. Furthermore, there is a unique measure $Q_{opt}$ which minimizes $Tr(V^Q)$.*

**Theorem 2.** *Suppose the pilot estimate of the influence function is consistent so that $\tilde{\psi}_\theta = \psi_{\theta_0} + o_p(1)$ under the uniform norm. Let $\hat{\pi}_n$ be estimated regularized inclusion probabilities for a sample of expected size $n_{sub}$ based on PPS sampling with size measure equal to the norm of the estimated influence function. As $n_{sub}, n \to \infty$ with $n_{sub}/n \to c > 0$, the plug-in estimator $\phi(\hat{\mathbb{P}}_n^{\hat{\pi}_n}) \rightsquigarrow \mathcal{N}(0, V^{Q_{opt}})$.*

We note this tight optimality result is the only one we are aware of as other methods only provide probabilistic bounds or rate guarantees save for the recent work in [27] specifically for logistic regression. Furthermore, probabilistic guarantees are immediately obtained as confidence intervals from our limit result, and these guarantees are tight.

Our optimality result is obtained assuming correct specification of the influence. Other methods may make fewer assumptions but at the cost of an optimality guarantee. These assumptions are often

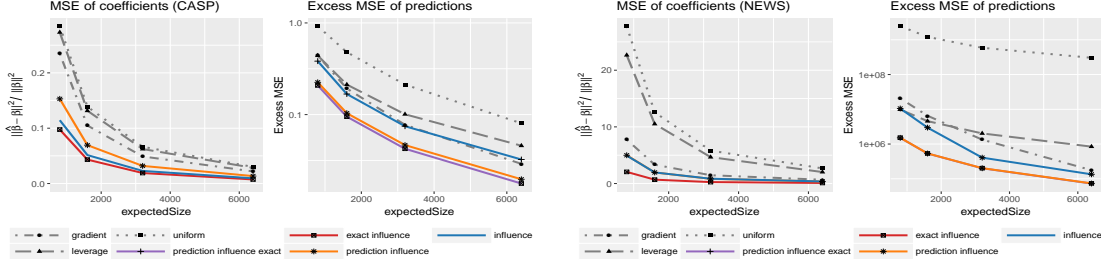

Figure 3: Results for linear regression on the CASP (left) and NEWS (right) data. Influence-based methods (colored) are top performers on CASP, requiring $\frac{1}{3}$ to $\frac{1}{2}$ the size of uniform sampling for the same error. As expected, the parameter influence sampling outperformed prediction influence sampling for parameter estimation and vice versa for prediction.

already made by the downstream model and, in some cases, are weak. For example, the squared error loss induces the same influence regardless of the underlying distribution. However, for quantile regression, the influence may depend on assumptions about how the error distribution changes with respect to the covariates $X$. This misspecification only affects the optimality and not correctness of the procedure. Even with a misspecified influence, an importance weighted subsample will still yield an unbiased estimate of the objective.

# 6   Experiments

We compared our sampling procedures with existing methods on three real datasets for least squares and quantile regression models. We did not include median L1 regression because it yields sampling with probability proportional to leverage which we did not consider to be an interesting comparison. Quantile regression, on the other hand, has a number of unique and applications beyond what can be found with a robust version of least squares: for instance, quality of service guarantees, understanding high-value donors for a nonprofit, or other cases where large values are of most interest.

The datasets we used are the CASP [18] ($n = 45730, d = 9$) and Online News Popularity ($n = 38644$, $d = 59$) datasets from the UCI repository, which are the same as those used by [29], and the EPA Risk-screening Environmental Indicators dataset [12] ($n \approx 9.6M, d = 10$), a much larger regression dataset. For the Online News Popularity dataset, we removed 4 columns due to collinearity. For the EPA dataset, we used $log(1 + Score)$ as the response. We expanded the 2 geographic coordinates into $10 \times 10$ cubic tensor b-splines and added log transformed predictors while removing the other $Score$ variables. This yielded a total of 111 regressors. In each case, we use a uniform random sample with size equal to the size of the smallest subsample considered to derive a pilot estimate and drew a weighted subsample from the remainder.

The quality of the fit on a subsample is measured either by either the relative squared error in the coefficients $\|\hat{\theta} - \theta_{opt}\|^2 / \|\theta_{opt}\|^2$ or by the mean excess loss $\mathbb{E}\,\ell(y_i, \hat{y}_i) - \mathbb{E}\,\ell(y_i, \hat{y}_i^{(opt)})$.

We performed experiments on both approximate influence and leverage scores as well as their exact counterparts. For the approximate methods, we use a Fast Walsh-Hadamard Transform to project $n$ rows to $10^4$ and compute the appropriate SVD or truncated SVD. We also consider two approximations to the influence: the diagonal approximation for the second moment matrix and the equal leverage approximation or, equivalently, the residual weighted approximation.

As shown in Figs. 3 and 4, our experiments demonstrate that influence based sampling outperforms all the other procedures. The improved performance depends both on the the design $X_{sub}$ and residuals (Fig. 1). When accounting for correlation in the design is costly simple approximations for the influence still yield significant improvements over uniform sampling (see supplementary materials).

Note that while we did not compare run times for these experiments, our technique should take the same time as leverage-based sampling, excluding time used to generate a pilot, as the only additional cost is computing residuals using the fixed pilot.

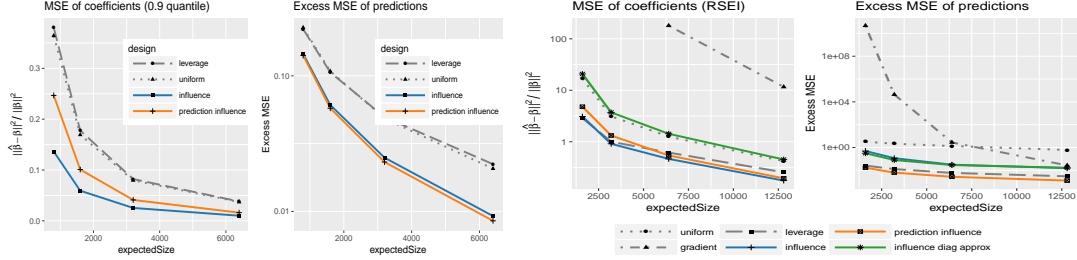

Figure 4: Results for a $90\%$-quantile regression model on the CASP data set (left) and linear regression on the large RSEI data set (right) also show approximate influence based sampling dominating other methods. Approximate leverage score sampling was slightly worse than uniform sampling on CASP and much worse on RSEI.

## 7    Discussion

One strength of sampling is the ability to reuse samples in a variety of problems. Since the samples are independent samples with known inclusion probabilities, they can be used in any procedure that takes importance weights. Multi-objective sampling [6] provides ways to sample for multiple objectives and generate a sample that is certifiably good across multiple tasks.

One particularly useful scenario for sampling is the case when a single machine can store and process the subsample. However, generating the sample required all the importance weights to first be computed and then iteratively normalized and regularized. To generate a sample on the fly, the adaptive threshold sampling technique in [25] can be used to both generate a fixed size, without replacement sample while ensuring a minimum sampling probability is satisfied.

Another consequence of work is that a reasonably good sample can be drawn as a side-effect of a stochastic gradient descent (SGD) procedure. A first-order SGD with no rescaling of the updates would generate gradient-based sampling weights, obviating the need for a special pilot with almost no additional cost. Computing just the residual as part of the SGD procedure or applying a diagonal approximation to the Hessian [3] would yield other natural approximations. In the case of a distributed SGD, the individual samples can be combined by a further round of subsampling to satisfy a storage constraint. If points are duplicated across machines, coordinated sampling [7] can be used to combine the samples.

We note that the regularization of probabilities is necessary to address inaccuracies from noisy pilot estimates, but the exact relationship is unknown. As such, we think theoretical analysis of the pilot and regularization would be interesting follow-on work.

## 8    Conclusion

We demonstrate both theoretically and empirically that influence functions yield good and principled subsampling procedures, achieving the best possible asymptotic variance over all independent, without replacement sampling designs with the same size and regularization. Our approach is also highly general as it applies to a wide range of statistical and machine learning models. Furthermore, it provides insight into the problem of optimal subsampling, allowing one to recast it as optimal sampling for a mean and decomposing the contributions of the regression design and response. This allows other existing work, such as on multi-objective sampling, to generate variations of this basic method that can address a wider range of scenarios. It further allows us to show several methods, including leverage score sampling, can be seen as approximations to our method. We also address computational difficulties with influence based methods and show fast approximations perform well.

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
