[Supplementary Material]

# Supplementary Materials

May 17, 2018

## A Proofs

**Theorem 1.** *For any $Q$ that satisfies the assumptions, $\sqrt{\frac{n}{c}}(\phi(\hat{\mathbb{P}}_n^Q) - \phi(P)) \rightsquigarrow Normal(0, V^Q)$ where $V^Q = \int \psi(x)\psi(x)^T \left(\frac{dP}{dQ}\right)^2 dQ(x)$. Furthermore, there is a unique measure $Q_{opt}$ which minimizes $Tr(V^Q)$.*

*Proof.* Since the weights $\frac{dP}{dQ}$ are bounded and fixed, $\sqrt{\frac{n}{c}}(\hat{\mathbb{P}}_n^Q - P) \rightsquigarrow R^Q$ where $R^Q$ is a tight process on $\ell(\mathcal{F})^\infty$, the space of functions in the Donsker class under the uniform norm. The functional delta method van der Vaart (2000) gives that $\sqrt{\frac{n}{c}}(\phi(\hat{\mathbb{P}}_n^Q) - \phi(P)) \rightsquigarrow \int \psi(x)dR^Q(x)$. Furthermore, this limit is $Normal(0, V^Q)$ where $V^Q = \int \psi(x)\psi(x)^T \left(\frac{dP}{dQ}\right)^2 dQ(x) = \int \psi(x)\psi(x)^T/\pi(x)dP(x)$ where $\pi = dQ/dP$.

Let $\Pi_{c,\alpha}$ be the set of measures on the sample space $\Omega$ that are absolutely continuous with respect to $P$ and such that for any $Q \in \Pi_{c,\alpha}$, the total measure $Q(\Omega) = c$ and $dQ/dP \geq \alpha$. It is easy to see that this set is closed and convex.

The optimization problem $\min_{Q \in \Pi_{c,\alpha}} Tr(V^Q)$ is thus a convex optimization problem having a strictly convex objective over a convex set of measures. Thus, it has a unique minimizer $Q_{opt}$. $\square$

**Theorem 2.** *Suppose the pilot estimate of the influence function is consistent so that $\tilde{\psi}_\theta = \psi_{\theta_0} + o_p(1)$ under the uniform norm. Let $\hat{\pi}_n$ be estimated regularized inclusion probabilities for a sample of expected size $n_{sub}$ based on PPS sampling with size measure equal to the norm of the estimated influence function. As $n_{sub}, n \to \infty$ with $n_{sub}/n \to c > 0$, the plug-in estimator $\phi(\hat{\mathbb{P}}_n(\hat{\pi}_n)) \rightsquigarrow Normal(0, V^{Q_{opt}})$.*

*Proof.* The distribution that attains the asymptotic variance $V^{Q_{opt}}$ is the one that samples with probability proportional to $\|\psi_{\theta_0}\|$ by the argument in section **??**. Thus, the only thing to prove is that the sample drawn using the estimated influences effectively yields the same sample as one using the exact influences.

The consistency of the estimated influence $\hat{\psi}$ gives that $\hat{\pi} = dQ/dP + o_p(1)$ for some constant $c$. For Poisson sampling with , the item $X_i, Y_i$ is in the sample if $U_i < \pi_i$ for independent $U_i \sim Uniform(0, 1)$. Let $\pi_i = dQ/dP(X_i)$ and $\tilde{\pi}_i$ be the estimated inclusion probability using the estimated influence. Let $Z_i = 1$ if $U_i < \pi_i$ and 0 otherwise. Likewise, $\tilde{Z}_i = 1$ if $U_i < \tilde{\pi}_i$ and 0 otherwise. Let $\epsilon_i = \tilde{\pi}_i - \pi_i$. The difference

$$Z_i/\pi_i - \tilde{Z}_i/\tilde{\pi}_i = \frac{Z_i - \tilde{Z}_i}{\pi_i}(1 - \epsilon_i + O(\epsilon_i^2)) \tag{1}$$

Taking the numerator, we have $|Z_i - \tilde{Z}_i| \sim Bernoulli(\epsilon_i)$. And the overall expectation of the absolute value is $O(\epsilon_i/\pi_i)$. Since $Var(Z_i/\pi_i) = (1 - \pi_i)$, it follows that the empirical estimate based on the estimated influences $\sqrt{\frac{n}{c}}(\hat{\mathbb{P}} - P)$ converges to the same limit as that under the optimal $Q_{opt}$. Hence, $\sqrt{m}(\phi(\hat{\mathbb{P}}) - \phi(P))$ and $\sqrt{m}(\phi(\hat{\mathbb{P}}^{Q_{opt}}) - \phi(P))$ also converge to the same limit by the functional delta method. $\square$

# B Additional figures for CASP dataset

## MSE of coefficients (CASP)

## Excess MSE of predictions

Figure 1: Results for simple to compute sampling weights show that even simple approximations lead to significant improvement over uniform sampling. Influence based and gradient based sampling performed similarly in this case. The residual approximation is the prediction influence weight with an equal leverage approximation.

# C Additional figures for NEWS dataset

## MSE of coefficients (0.9 quantile)

Figure 2: Results for 90% quantile regression on the NEWS dataset.

# D    Additional figures for simulated dataset

Figure 3: Results for linear regression on synthetic data. Gradient based sampling is nearly the same as prediction influence when the regressor columns are i.i.d.

# References

van der Vaart, A.W. *Asymptotic Statistics*. Cambridge University Press, 2000.