[Reviews · NeurIPS 2018]

Reviewer 1



The paper presents a sampling methodology for statistical estimation tasks, in which the weight of a sample (to be included in final estimation) is determined by the sample's Gateaux derivative. The authors point out that Gateaux derivatives are defined for several estimators of note, and that asymptotically characterize estimate errors for so-called asymptotically linear estimators, which is also a very broad class. The authors then propose a probabilistic sampling method in which samples are included with a probability proportional to the norm of the gateaux derivative, and show that this is optimal, in that it minimizes the trace of the variance of the estimator under certain conditions. They then proceed to apply this method to least squares estimation, GLMs, and quantile regression, on all of which estimation leads to improved performance (both in estimation and prediction) compared to alternative methods for subsampling. This is an interesting, solid contribution, that addresses a fundamental problem. On the negative side, the paper is unpolished/somewhat written in a rush, and is not appropriately positioned with respect to related work. This seems a significant drawback when dealing with a topic as fundamental as sampling. To begin with, the related work (there is no dedicated section) is very sparse, citing mostly TCS papers; what is the state of the art in sampling in statistics literature? Have gateaux derivatives been used before for this purpose? How does the present work relate to other uses of gateaux derivatives in statistics and estimation in particular? On a similar vein, the paper mentions several results, presumably already known without any citations or context whatsoever. Asymptotically linear estimators are an example. Clearly, gateaux derivatives for the specific estimation tasks in Section 4 must also have been derived before, and although the discussion of the resulting sampling methods is interesting, again credit for this is due. Similarly, competing sampling methods are sometimes vaguely disparaged as "only provid(ing) probabilistic or rate guarantees". What are the competing methods, and what is the exact formulation of their probabilistic and rate guarantees? Given the broad array of estimators termed to be asymptotically optimal, one would expect more examples listed and evaluated (beyond the usual LSE and GLS). The same holds for competing sampling methods. Finally, the theoretical statement (Theorems 1 and 2) and their preamble are too dense, making it difficult to assess the breadth and applicability of the conditions required for the theorems to hold. ---Added post-rebuttal--- I appreciate the authors' detailed response, and the explanation of the lack of work in statistics on subsampling. If some of these comments could make it to the paper, that would be great. --------------------------------

Reviewer 2



[post rebuttal] The author is able to give constructive comments based on the feedback of reviewers. [A summary of the paper (motivation, methods, results) written in your own words of about a paragraph.] The authors developed a novel influence-function (IF) based sampling method that is asymptotically optimal and can be applied to a wide range of parameters (tasks), namely asymptotically linear estimators, which fills the gap between efficient empirical process theory and computational statistics. The authors presented the canonical formulation of IF based sampling as well as approximated methods for computation efficiency, and illustrated their method in three examples (OLS, GLM, quantile regression), and for each example presented influence functions for the \theta parameter and the conditional mean E(Y|X) parameter. In the examples, the authors established connection between (1) leverage score based sampling and (2) the IF based sampling for prediction \hat y_i. Connections between localcase-controlsamplg and IFbasedsamplgforGLMprediction is also presented. The authors proved asymptotic optimality of their proposed method. The authors conducted simulation study to evaluate the performance of the new method compared to current state-of-the-art sampling methods. On three datasets, the authors run OLS and 90% quantile regression, and evaluate based on two metrics. And shows that IF-based sampling reduces MSE for both prediction and parameter estimation. [A list of major issues] - in section 6 (experiments), the authors implemented the method for 90%quantiregression rather than medianregression, which is the most widely applied method. It would make the paper much more applicable and useful for practitioners if the medianregression method is also evaluated in simulation [A list of minor issues] NA

Reviewer 3



Main Idea: This paper proposes a new subsampling algorithm for a wide range of models. The authors show that the optimal subsampling problem can be reduced to subsampling of univariate mean by utilizing influence functions. The authors show that the proposed method is asymptotically optimal in terms of variance. The authors show how the approach can be used to several popular models including linear regression, quantile regression and generalized linear model. Experiments on three real-world dataset demonstrates the superiority compared to state-of-the-art baselines. Strength: 1. The authors provide an interesting and important insight that optimal subsampling can be reduced to subsampling of mean by utilizing influence function. This finding is innovative. 2. The proposed method has wide applications to several popular models. 3. The authors provide both theoretical optimality guarantee and empirical evidence on the superiority of the proposed method. Weakness: 1. The authors describe methods for accelerating influence function computation. It is better if the authors can include comparison of running time for the proposed method and the baselines in the experiments. 2. The proposed method requires a pilot estimation of the parameter. It is not clear how the pilot estimation is carried out in the experiments. Also, it is interesting to see how the pilot estimation accuracy impacts the performance of the subsampling algorithm. Detailed comments: 1. Line 97, missing space in For example[11, 14]. 2. Line 242, almost everywhere -> exists almost everywhere.